# Identification of Sources and Transformations of Nitrate in the Intense Human Activity Region of North China Using a Multi-Isotope and Bayesian Model

**DOI:** 10.3390/ijerph18168642

**Published:** 2021-08-16

**Authors:** Chaobin Ren, Qianqian Zhang, Huiwei Wang, Yan Wang

**Affiliations:** 1School of Chemical Engineering, Zhengzhou University, Zhengzhou 450001, China; robin_ren@gs.zzu.edu.cn (C.R.); wangyan371@zzu.edu.cn (Y.W.); 2School of Civil Engineering, Nanyang Institute of Technology, Nanyang 473004, China; 3Institute of Hydrogeology and Environmental Geology, Chinese Academy of Geological Sciences, Shijiazhuang 050061, China; whuiwei@mail.cgs.gov.cn; 4Key Laboratory of Cenozoic Geology and Environment, Institute of Geology and Geophysics, Chinese Academy of Sciences, Beijing 100029, China

**Keywords:** nitrate, pollution sources, isotope, source apportionment, Bayesian model

## Abstract

Nitrate (NO_3_^−^) contamination in water is an environmental problem of widespread concern. In this study, we combined the stable isotopes of NO_3_^−^ (δ^15^N and δ^18^O) and water (δ^2^H and δ^18^O) with a Bayesian mixing model (SIAR) to identify the sources and transformation of NO_3_^−^ in groundwater and rivers in the Ye River basin of North China. The results showed that the mean NO_3_^−^ concentrations in groundwater were 133.5 and 111.7 mg/L in the dry and flood seasons, respectively, which exceeded the required Chinese drinking water standards for groundwater (88.6 mg/L) (GB14848-2017). This suggests that groundwater quality has been severely impacted by human activity. Land use significantly affected the concentration of NO_3_^−^ in the Ye River basin (*p* < 0.05). However, the NO_3_^−^ concentrations in groundwater and river water had no obvious temporal variation (*p* > 0.05). The principal mode of nitrogen transformation for both groundwater and river water was nitrification, whereas denitrification did not significantly affect the isotopic compositions of NO_3_^−^. The sources of NO_3_^−^ mainly originated from sewage and manure, soil nitrogen, and NH_4_^+^ in fertilizer for groundwater and from sewage and manure for the river water. According to the SIAR model, the primary sources of nitrate found in groundwater and river were sewage and manure in the Ye River basin. The proportional contributions of sewage and manure to nitrate contamination of groundwater and river were 58% and 48% in the dry season and 49% and 54% in the flood season, respectively. Based on these results, we suggest that the local government should enhance the sewage treatment infrastructure, construct an effective waste storage system to collect manure, and pursue a scientific fertilization strategy (such as soil formula fertilization) to increase the utilization rate of nitrogen fertilizer and prevent nitrate levels from increasing further.

## 1. Introduction

Nitrate pollution in water environments has become an important problem of worldwide concern [1,2]. High nitrate content in water is closely related to human activities such as the excessive use of nitrogen fertilizers [3,4], the discharge of unprocessed domestic sewage and industrial wastewater [5], sewage irrigation [6], landfill seepage, and atmospheric N deposition [7].

Nitrate entering an aquifer can remain stable for decades in an oxidizing condition due to its highly solubility and stability, and it migrates readily [6]. Excessively high concentrations of nitrate in drinking water can pose serious threats to human health (e.g., methemeoglobinemia in infants, thyroid disorders, and cancer of the digestive tract) [8,9] and aquatic ecosystems (e.g., eutrophication and seasonal hypoxia) [10]. Not surprisingly, the World Health Organization (WHO) has responded to these conditions by setting a guideline for the upper value for nitrate in drinking water at 50 mg/L [11]. Therefore, it is extremely important to accurately identify sources of nitrate and prevent nitrate concentrations from continuously increasing.

Nitrate in the water environment usually originates from multiple sources [9,12] such as sewage and manure (SAM), soil nitrogen (SN), NO_3_^−^ fertilizer (NF), NO_3_^−^ in precipitation (NP), and NH_4_^+^ in fertilizer and rain (NFAR) [13,14,15]. Therefore, it is difficult to precisely distinguish among these sources of nitrate via hydrochemistry methods alone. A dual isotope (δ^15^N-NO_3_^−^ and δ^18^O-NO_3_^−^) method has proven to be a powerful indicator for accurately identifying nitrate sources because isotope values of nitrate from different sources are different [5,16,17,18,19]. Previous studies have found that the typical δ^15^N values of precipitation, soil nitrogen, and chemical fertilizers ranges from −13‰ to +13‰, from −3‰ to +9‰, and from −6‰ to +6‰, respectively [4,14]. Manure and sewage are enriched in ^15^N relative to other sources, and so, they have higher δ^15^N values, spanning from +4‰ to +25‰ [20]. It can be seen that the ranges of δ^15^N-NO_3_^−^ from different sources are partly overlapped. However, the δ^18^O-NO_3_^−^ values can provide useful information for discriminating among precipitation (from +25‰ to +75‰), chemical fertilizers (from +17‰ to +25‰), and nitrification (including soil nitrogen, manure and sewage, fertilizers, etc.) (from −10‰ to +10‰) due to its distinct isotopic signatures [8].

Nevertheless, during nitrogen’s transport in water and soil, the original nitrate isotope composition of different sources can change because of isotopic fractionation during biogeochemical processes such as nitrification and denitrification [21,22]. This limitation may affect the accuracy of tracing sources of nitrates. Some researchers, accordingly, have combined the use of the dual isotopic of nitrate with other isotopes (such as δ^18^O-H_2_O, δ^37^Cl, δ^11^B) [2,20,23], statistical methods (such as principal component analysis) [24] and hydrochemical characteristics (such as NO_3_^−^/Cl^−^, I/Na vs. Br^−^) [25] to identify the pollution source more accurately.

In addition, quantifying the relative contribution of different sources is critical for developing a priority strategy for controlling the nitrate contamination of water. At present, the main quantitative source analysis model is the Bayesian mixing model (i.e., stable isotope analysis in R or SIAR). The model not only takes into account the effect of isotopic fractionations on sources’ apportionment, but also can trace more than three sources [4,26]. In recent years, the SIAR model has been successfully applied to quantify the proportional contributions of the different sources of nitrate in water. However, there are few studies on nitrate transformation and source apportionment in the surface- and groundwater of river belts, where human activities have a strong impact on the water environment.

The Ye River is a tributary of the Hutuo River located in the southwestern Hebei Province, China. The Ye River, a mountain river that originates from the Miao River in Shouyang County of Shanxi Province, flows from the southwest to the northeast and empties into the Huangbizhuang Reservoir in Pingshan County. Groundwater in this area is the main supply of industrial, agricultural, and drinking water. In the Ye River basin, the main land use type is agricultural land, the villages are mainly distributed on both sides of the river, and most villages do not have sewerage systems. Therefore, groundwater and river water could be affected by a variety of pollutants. Previous research found that groundwater quality had deteriorated because of human activities, and the concentration of groundwater nitrate exceeded the threshold value for drinking water set by the WHO (50 mg/L) [27,28,29]. However, surprisingly, few studies have investigated both the sources and the fate of nitrate in this region at the basin scale.

In this study, we distinguished the characteristics of the spatial and seasonal variations in nitrate in groundwater and river water in an intense human activity region. Nitrate sources across different seasons and the transformations of nitrogen were identified by combining multiple isotopes (δ^15^N-NO_3_^−^, δ^18^O-NO_3_^−^, δ^2^H-H_2_O, and δ^18^O-H_2_O) and hydrochemical characteristics. Proportional contributions of multiple nitrate sources in the groundwater and river water were estimated using a Bayesian isotopic mixing model. Collectively, these results are helpful for elucidating the underlying mechanism of nitrate pollution and for developing effective water quality protection strategies for this and other mountain river basins with differing anthropogenic influences.

## 2. Materials and Methods

### 2.1. Study Area

The Ye River is located in the North China Plain. The study area spans from Yangquan County (Shanxi Province) to the Huangbizhuang Reservoir in Pingshan County (Hebei Province), with a total population of about 1 million. The topography of the Ye River basin inclines from southwest to northeast, and this basin covers approximately 600 km^2^. The study region has a semi-arid monsoon climate, with an average annual precipitation of 450–750 mm (mostly falling in June to September) and a mean annual temperature between 17–29 °C [14]. The main types of land use here include agricultural land (39.1%), forest land (41.3%), lawn land (13.2%), and construction land (5.2%) along with surface water bodies (1.2%) (Figure 1).

The main aquifer is part of the Quaternary multi-layer aquifer system of the Hebei Plain [2]. Its lithology consists of gravel, pebbles, coarse sand, and fine sand (Appendix A). The principal types of groundwater in this basin are fissure water and loose-stratum pore water. The aquifer in the study area is rich in water, and the flow direction of groundwater is from southwest to northeast. The groundwater is recharged mainly by precipitation, river inputs, and irrigation return. Its discharge mode occurs primarily via manual exploitation. The depth of the groundwater table is 2–35 m.

### 2.2. Sample Collection and Analysis

Water samples were obtained from the Ye River basin in April 2018 (dry season) and August 2018 (flood season). 25 samples were collected including 16 groundwater samples (G01–G16), 6 surface water samples, and 3 sewage samples (Figure 1). In all, 13 rain events were monitored (during April–October 2018), and the sampling site (longitude: 114°28′30.49″; latitude: 38°5′3.52″) was located on the rooftop of the institute in Shijiazhuang City, China. The groundwater was mainly collected from civil wells and agricultural irrigation wells. The depth range of sampling wells was 6.0–60 m.

Groundwater samples were extracted by pumping water from the wells, and wells were flushed approximately 20 min before sampling until the pH and EC of the water were stable. The river water samples were collected in the middle of the river from about 30 cm below the water surface by using a surface water sampler made of plexiglass. The sampling of groundwater and river water was completed within two days. Rain samples were collected in a simple, self-made water-collecting device. The structure of the device was as follows: A polyethylene bucket (diameter: 0.5 m; height: 1 m) was selected as a rainwater collecting frame, a funnel-shaped polyethylene film covered the top of the rainwater collecting frame, and a 1 L water-collecting bottle was placed below the film. In addition, the bottle was connected to a funnel with a diameter of 25 cm, and a table tennis ball was placed inside the funnel to prevent the rain from evaporating. Rainwater samples were collected as soon as the rain stopped. The values of the pH, dissolved oxygen (DO), and electrical conductivity (EC) were measured in the field using a multiparameter instrument (HACH HQ40d, Loveland, CO, USA). All water samples were filtered through 0.45-μm membrane filters in the lab and then stored in high-density polyethylene bottles for later analysis of their main ions, namely nitrate (NO_3_^−^), nitrite (NO_2_^−^), ammonia (NH_4_^+^), chloride (Cl^−^), and manganese (Mn) as well as four signature isotopes, namely δ^15^N-NO_3_^−^, δ^18^O-NO_3_^−^, δ^2^H-H_2_O, and δ^18^O-H_2_O.

NO_3_^−^, NO_2_^−^, and NH_4_^+^ were measured using spectrophotometry (PerkinElmer Lambda 35, Waltham, MA, USA) with a precision of ±5%. NO_3_^−^ and NO_2_^−^ were determined by using the phenol disulfonic acid spectrophotometry method and the a-naphthylamine spectrophotometry method, respectively, while NH_4_^+^ was analyzed using Nessler’s reagent spectrophotometry method. Cl^−^ was measured using an ion chromatograph (Dionex, Sunnyvale, CA, USA, ICS-1000). Mn was measured using inductively coupled plasma mass spectrometry (Agilent 7500ce ICP-MS, Tokyo, Japan). Hydrochemistry parameters were analyzed in the laboratory of the Groundwater Mineral Water and Environmental Monitoring Center at the Institute of Hydrogeology and Environmental Geology of the Chinese Academy of Geological Sciences.

The values of δ^15^N-NO_3_^−^ and δ^18^O-NO_3_^−^ in the samples of groundwater, river water, and rainwater were determined using denitrifier methods [30] with an isotope ratio mass spectrometer (Delta V Plus, Finnigan, Bremen, Germany) at the Institute of Environment and Sustainable Development in Agriculture, Chinese Academy of Agricultural Sciences, Beijing City, China. The values of δ^2^H-H_2_O and δ^18^O-H_2_O were analyzed by a Finnigan MAT 253 mass spectrometer at the Institute of Hydrogeology and Environmental Geology, Chinese Academy of Geological Sciences, Beijing City, China. For these obtained δ^15^N-NO_3_^−^, δ^18^O-NO_3_^−^, δ^2^H-H_2_O, and δ^18^O-H_2_O values, their analytical precision was ±0.2‰, ±0.25‰, ±0.1‰, and ±0.5‰, respectively. All stable isotope ratios were expressed in per mil (‰) deviations as follows:(1)$$\delta_{\mathrm{sample}}(‰)=(\frac{R_{\mathrm{sample}}}{R_{\mathrm{standard}}}-1)\times 1000$$
where R_sample_ and R_standard_ are the ^15^N/^14^N or ^18^O/^16^O or ^2^H/^1^H ratios of the samples and standards, respectively. The isotopic values are reported relative to N_2_ in atmospheric air (AIR) for δ^15^N and to Vienna Standard Mean Ocean Water (VSMOW) for both δ^18^O and δ^2^H [31].

### 2.3. Bayesian Mixing Model (SIAR)

The contribution of a given nitrate source to the total nitrate in groundwater or river water was expressed as a proportion. This was determined using the Bayesian isotopic mixing model implemented in the “SIAR” (Stable Isotope Analysis in R) software package. Detailed instructions for the model can be found in Parnell et al. (2010) [32].

### 2.4. Multivariate Statistical Analysis

One way analysis of variance (ANOVA) was used to distinguish the differences in the NO_3_^−^ concentration and the δ^15^N-NO_3_^−^, δ^18^O-NO_3_^−^, δD, and δ^18^O-H_2_O values among the different regions and seasons. The ANOVAs were implemented in SPSS software (v. 21.0; SPSS, Inc., Chicago, IL, USA).

## 3. Results

### 3.1. Water Chemistry Characteristics

The pH, EC, and DO values of the samples are displayed in Table 1. As shown in Table 1, pH was neutral to mildly alkaline (6.91 to 7.72 and 6.98 to 8.35 in the dry and flood seasons, averaging 7.36 and 7.44, respectively). However, the mean pH values of river water (8.32 and 8.27 in the dry and flood seasons, respectively) were higher than those of groundwater, which may have been because the evaporation of surface water was stronger than that of the groundwater [14]. Its EC values varied from 886 to 1986 μs/cm in the dry season and 807 to 2260 μs/cm in the flood season with mean values of 1293 μs/cm and 1354 μs/cm, respectively. For river water, in contrast, the mean EC value in the flood season (1058 μs/cm) was lower than that in the dry season (1258 μs/cm). The DO level in the water environment is an important indicator for gauging its redox condition and the process of nitrogen transformation. Groundwater DO varied from 2.67 to 8.79 mg/L (mean: 6.46) in the dry season and from 2.95 to 9.45 mg/L (mean: 6.86) in the flood season. For river water, the DO concentration was 9.18 and 8.22 mg/L in the dry and flood seasons, respectively, and evidently higher than that of groundwater.

### 3.2. Temporal and Spatial Variation of Nitrogenous Compounds

The river NH_4_^+^ varied from BDL to 0.59 mg/L (mean: 0.24 mg/L) in the dry season and from BDL to 0.59 mg/L (mean: 0.17 mg/L) in the flood season. The concentration of NH_4_^+^ in groundwater samples was below the detection limit (BDL: 0.04 mg/L). All these cases met the required Chinese drinking water standards for surface water (i.e., 1.29 mg/L) [33]. The NO_2_^−^ concentration in groundwater ranged from BDL (0.002 mg/L) to 0.021 mg/L (mean: 0.006) in the dry season and from BDL to 0.024 mg/L (mean: 0.007) in the flood season. Here, too, all cases met the required Chinese drinking water standards for groundwater (i.e., 3.29 mg/L) [34]. Though the mean concentrations of NO_2_^−^ (0.228 and 0.191 mg/L in the dry and flood seasons, respectively) were higher in river water than groundwater (0.006 and 0.007 mg/L in the dry and flood seasons, respectively), there were no clear trends in the spatial and temporal variation of NO_2_^−^ in either groundwater or river water.

The temporal and spatial variation of NO_3_^−^ in groundwater and river water is depicted in Figure 2. The NO_3_^−^ concentration in groundwater ranged from 15.1 to 376.5 mg/L (mean: 133.5) in the dry season and from 16.7 to 273.1 mg/L (mean: 111.7) in the flood season. NO_3_^−^ was the primary nitrogenous species in the Ye River basin, and its concentration was significantly higher than either NH_4_^+^ or NO_2_^−^. However, the mean concentration of NO_3_^−^ exceeded the required Chinese drinking water standards for groundwater (i.e., 88.6 mg/L) [34]. In contrast, the NO_3_^−^ concentration in river water, which ranged from 17.4 to 27.9 mg/L (mean: 23.7) in the dry season and from 21.1 to 26.7 mg/L (mean: 20.0) in the flood season, met the required Chinese drinking water standards for surface water (44.3 mg/L) [33]. As shown in Figure 2a, there was no significant temporal variation in the NO_3_^−^ concentration in groundwater or river water (*p* > 0.05). However, the land use type significantly influenced the concentration of NO_3_^−^ in the Ye River basin. As can be seen in Figure 2b, the mean concentration of groundwater NO_3_^−^ was significantly higher in the agriculture area (215.9 mg/L) than in the county area (103.9 mg/L), the village area (76.1 mg/L), or the river (23.9 mg/L) (*p* < 0.05).

### 3.3. Spatial and Temporal Variation of Stable Isotopes

#### 3.3.1. Variation in the δD and δ^18^O Values of Groundwater, River Water, and Rainfall

Figure 3 shows the temporal variation of δD-H_2_O and δ^18^O-H_2_O values in rainfall, groundwater, and river water. In rainfall, δD-H_2_O and δ^18^O-H_2_O ranged from −100‰ to −38.0‰ and from −14.6‰ to −6.0‰ (means: −63.1‰ and −9.4‰), respectively. In groundwater, δD-H_2_O and δ^18^O-H_2_O respectively ranged from −70.3‰ to −60.1‰ and −10.2‰ to −8.8‰ (corresponding means: −69.3‰ and −9.5‰) in the dry season and from −73.2‰ to −66.2‰ and −10.8‰ to −9.8‰ in the flood season (corresponding means: −66.7‰ and −10.2‰). In river water, the δD-H_2_O and δ^18^O-H_2_O ranged from −68.4‰ to −62.7‰ and −10.0‰ to −9.3‰ (means: −65.4‰ and −9.6‰) in the dry season, respectively, and likewise from −71.1‰ to −68.0‰ and from −10.5‰ to −10.3‰ in the flood season (corresponding means: −70.1‰ and −10.4‰). Both δD-H_2_O and δ^18^O-H_2_O in groundwater and river water in the dry season were significantly higher than in the flood season (*p* < 0.05; Figure 3). However, no spatial differences were detected for δD-H_2_O and δ^18^O-H_2_O in groundwater and river water (*p* > 0.05).

#### 3.3.2. Variation in the δ^15^N-NO_3_^−^ and δ^18^O-NO_3_^−^ Values of Groundwater and River Water

As shown in Figure 4a,b, the δ^15^N-NO_3_^−^ values in the groundwater of the Ye River basin varied from 6.27‰ to 14.56‰ and 4.85‰ to 12.10‰ in the dry and flood seasons, averaging 8.85‰ and 7.51‰, respectively (Figure 4a). The δ^18^O-NO_3_^−^ varied from 1.23‰ to 13.34‰ and from −1.40‰ to 10.97‰ in the dry and flood seasons, respectively, with corresponding averages of 6.84‰ and 5.30‰ (Figure 4b). Yet, no significant temporal and spatial variation was detected for the δ^15^N-NO_3_^−^ and δ^18^O-NO_3_^−^ in groundwater (*p* > 0.05; Figure 4c,d). In river water, the δ^15^N-NO_3_^−^ and δ^18^O-NO_3_^−^ had means of 10.16‰ and −2.07‰ and 11.13‰ and 3.65‰ in the dry and flood seasons, respectively, but showed no significant temporal variation (*p* > 0.05; Figure 4a,b). However, the mean δ^15^N-NO_3_^−^ value was significantly higher in river water than that in groundwater (Figure 4c), whereas this was reversed for δ^18^O-NO_3_^−^ (Figure 4d), a result pointing to differential N sources.

## 4. Discussion

### 4.1. Water Origin and Recharge

In this study region, the regression lines of the δD and δ^18^O in groundwater and river water samples were close to the global and local meteoric water lines (GMWLs and LMWLs), respectively (Figure 5) (GMWL, defined as δD = 8 δ^18^O + 10; LMWL, δD = 7.25 δ^18^O + 4.85; R^2^ = 0.90). In addition, most of the isotopic composition points fell below the GMWL, suggesting that rainfall underwent little evaporation before recharging the aquifer and the river [21,35]. It is worth noting that the regression line slopes for groundwater and river water were similar, and the isotopic composition of groundwater was close to the river water, which indicated that the groundwater was affected by the seepage of the Ye River [35,36]). In the Ye River basin, the stratum lithology is mainly coarse and sandy gravel; therefore, the surface water is easy to infiltrate and recharges the groundwater. In addition, the table of groundwater was deep (3–22 m), and there was no spring outlet in the study area; thus, their conversion relationship was mainly the river recharging the groundwater. Therefore, the accumulating nitrogen pollutants in river water increased the NO_3_^−^ concentration of groundwater.

### 4.2. Identification of Nitrate Sources Using Stable Isotope Methods

As shown in Figure 6, the δ^15^N and δ^18^O values of groundwater fell within the ranges of NFAR, SN, and SAM. Hence, the groundwater NO_3_^−^ in this region may have multiple sources. Indeed, in the Ye River basin, the main land use type is agricultural land (39.1%). Chemical fertilizers could be important sources of NO_3_^−^, and we found that nitrogen fertilizers (such as urea, ammonium salts) were the primary agricultural inputs used in our study region. In addition, manure is also often used in farmland. According to previous studies, the utilization rate of chemical fertilizer by plants was 61–65%, leaving the remainder lost to the environment [37]. Among the groundwater samples, 15% and 100% of them fell within the ranges of the δ^15^N and δ^18^O values for the NH_4_^+^ fertilizer and manure, respectively. Therefore, the NH_4_^+^ fertilizer and manure could be principal sources of NO_3_^−^. Previous research found that the δ^15^N and δ^18^O values for NO_3_^−^ fertilizer, NH_4_^+^ fertilizer, and sewage and manure could range from −6‰ to +6‰, −4‰ to +6‰, and +4‰ to +25‰ and from +17‰ to +25‰, −10‰ to +10‰, and −10‰ to +10‰, respectively [8,38,39]. In the Ye River basin, NF is less used, and the δ^15^N and δ^18^O values were not within the range of NF. Thus, it may not be a key source of NO_3_^−^.

Sewage is the other important pollution source of groundwater NO_3_^−^ in the Ye River basin. During the study period, we found that the domestic sewage of most villages drained directly into the Ye river. As noted in the preceding paragraph, all groundwater samples were within the range of the δ^15^N and δ^18^O values for sewage and manure (respectively, from +4‰ to +25‰ and −10‰ to +10‰) [2]. In addition, the mean concentration of NO_3_^−^ from sewage samples (n = 3) was 55.2 mg/L, and this had a large impact on groundwater nitrate pollution. Thus, we deduce that sewage is also a major pollution source of NO_3_^−^ in the basin.

Nitrate in the water environment originating from soil nitrogen is a product of the bacterial decomposition of organic materials [40]. In this study, soil nitrogen may have some influence on the nitrate pollution of groundwater. 59% of samplings of groundwater fell within the overlap ranges of the δ^15^N and δ^18^O values for the SN (δ^15^N: from −3‰ to +9‰; δ^18^O: from −10‰ to +10‰) [4] and SAM. Furthermore, the stratum lithology is mainly coarse and sandy gravel, and as a result, soil nitrogen can easily infiltrate into groundwater. Therefore, soil nitrogen is also a pollution source of groundwater NO_3_^−^.

Precipitation may not be a main source of groundwater NO_3_^−^ in the Ye River basin because the δ^15^N and δ^18^O values of groundwater were not within the ranges (from −1.0‰ to +6.4‰ and +28.9‰ to +66.4‰, respectively) of rain samples collected during the study period. In addition, the mean concentration of NO_3_^−^ in the rainfall samples (n = 13) was 11.1 mg/L, which had little impact on groundwater nitrate pollution (corresponding means were 133.5 mg/L and 111.7 mg/L in the dry and flood seasons, respectively). As a result, precipitation was not a primary source of groundwater NO_3_^−^ in this region.

In comparison with groundwater, the δ^15^N and δ^18^O values of the river water were more concentrated and fell within the ranges of SAM (Figure 6). Thus, SAM was likely a predominant source of river NO_3_^−^ in this region, which was consistent with the aforementioned drainage into nearby rivers of the domestic sewage of most villages. Further, manure does not easily dissolve after being applied to soil, so it could be carried by rainfall runoff to enter river water and increase the NO_3_^−^ concentration of the latter. Therefore, we can be sure that SAM constitutes the main source of NO_3_^−^ in river water. Chemical fertilizer could have a slight impact on river NO_3_^−^ contamination. This is related to the local fertilization and irrigation ways. Normally, once fertilizer is applied to the farmland, irrigation is carried out immediately (the main method of irrigation is flooding), which causes the fertilizer unused by plants to migrate into groundwater. Thus, chemical fertilizer may not be a main source of river water NO_3_^−^. Precipitation cannot be a primary source of NO_3_^−^ in river water because its δ^15^N and δ^18^O values were well out of the range reported for precipitation.

### 4.3. Transformation of Nitrogen

Nitrification and denitrification are the two most important processes for nitrogen transformation because they can affect the original δ^15^N and δ^18^O values of nitrate via isotopic fractionation [21]. Theoretically, the δ^18^O-NO_3_^−^ produced by microbial nitrification is calculable using the formula δ^18^O-NO_3_^−^ = 1/3 δ^18^O–O_2_ + 2/3 δ^18^O–H_2_O, in that one-third of the oxygen in NO_3_^−^ should be derived from oxygen in the air (+23.5‰) [41], while two-thirds should come from ambient water (from −10.8‰ to −8.8‰ and from −10.5‰ to −9.3‰ for groundwater and river water, respectively) at the site of nitrate formation [2]. As shown in Figure 7, most of the δ^18^O-NO_3_^−^ values in groundwater and river water were higher than the theoretical range of δ^18^O-NO_3_^−^ derived from nitrification, which might have been related to the evaporation of both soil water and river water, or it might have arisen due to the higher δ^18^O–O_2_ caused by bacterial respiration [14]. Previous studies have found that the nitrate produced from nitrification processes has a range of δ^18^O-NO_3_^−^ values, spanning from −10‰ to +10‰ [41]. In our study, the δ^18^O-NO_3_^−^ values for 81% of the groundwater and 92% of the river water samples fell within the typical range of nitrification, indicating that it was the main process of nitrogen transformation in the Ye River basin.

Denitrification is vital for removing nitrate from a polluted water environment, and this process is more likely to occur when oxygen is limited, but organic carbon is available [13]. Our results suggest that denitrification is probably not a main process for nitrogen transformation in the basin given the higher concentration of DO (for groundwater and river water, DO varied from 2.67 to 9.45 mg/L and from 7.33 to 10.78 mg/L, respectively), which was not within the suitable denitrification oxygen level (<2.0 mg/L) [4]. Other research has found that denitrification increased the δ^15^N-NO_3_^−^ and δ^18^O-NO_3_^−^ values of water samples as the NO_3_^−^ concentration decreased to yield δ^15^N-NO_3_^−^ and δ^18^O-NO_3_^−^ ratios of 1.3–2.1:1 [15,42]. As can be seen in Figure 8, no significant negative correlation was present between NO_3_^−^ concentrations and the values of δ^15^N-NO_3_^−^ or δ^18^O-NO_3_^−^. Therefore, denitrification was not a primary process for nitrogen transformation in the Ye River basin.

In addition, assimilation is also an important process for nitrogen transformation in streams [43]. Previous studies have found that algae/biology generally prefers to uptake light isotopes of nitrate, which leads to the enrichment of heavy isotopes in residual nitrate [17] and causes a decrease in nitrate concentration [44]. In this study, the concentration of nitrate did not decrease gradually, and the isotope values of δ^15^N-NO_3_^−^ and δ^18^O-NO_3_^−^did not increase gradually from upstream to downstream (Appendix A). Therefore, assimilation by algae/biology may not be a main process of nitrogen transformation in the Ye River.

### 4.4. Estimation of Proportional Contributions from Dominant Nitrate Sources Using a SIAR Model

A Bayesian mixing model (SIAR) was used to evaluate the proportional contributions of nitrate from five potential nitrate sources: SAM, SN, NF, NP, and NFAR. We collected the values of δ^15^N-NO_3_^−^ and δ^18^O-NO_3_^−^ from the literature (reviewed in Zhang et al. 2015) [31] and measured the values for precipitation in our study region. The fractionation factor C_jk_ was presumed to be 0, as denitrification had less effect on the isotopic compositions of groundwater and river water.

#### 4.4.1. Estimates of Proportional Contributions of Nitrate Sources in Different Seasons

As Figure 9a shows, the primary nitrate sources in groundwater and river water were SAM in Ye River basin. The proportional contribution of SAM in groundwater was slightly higher in the dry season (58%) than in the flood season (49%) because sewage was diluted by rainfall in the flood season. However, the proportional contribution of SAM to river water contamination showed that this was slightly higher in the flood season (54%) than in the dry season (48%), possibly because rainfall runoff carried the manure that was piled on the road and the sewage left in the river course entered the river. Soil nitrogen was the second major source of NO_3_^−^, contributing to groundwater at 22% and 25% in the dry and flood seasons, respectively, and correspondingly, at 19% and 18% to river water.

The contribution proportion from atmospheric precipitation to the NO_3_^−^ in the river (9% and 5% in the dry and flood seasons, respectively) was slightly higher than that in the groundwater (corresponding values: 1% and 2%). This could be explained by rainfall being a direct recharge to river water, and the NO_3_^−^ concentration of river water was relatively low. This result suggests that the NO_3_^−^ in river water is easily affected by precipitation regimes. The contribution from NH_4_^+^ fertilizer (15%) was higher than that from NO_3_^−^ fertilizer (7%) because the latter is not commonly used in China [17].

Based on the SIAR model results, point sources (sewage and manure) are still the most crucial sources of nitrate in groundwater and river water in this mountain river basin of China. Therefore, local governments should take action to strengthen the sewage treatment infrastructure, build a waste storage system to collect the manure, and also pass strict legislation to prohibit the haphazard heaping of manure by roadsides. Additionally, local government should pursue a scientific fertilization strategy such as soil formula fertilization in order to increase the utilization rate of nitrogen fertilizer. Implementing the above-mentioned measures in a timely way can prevent the nitrate levels in the Ye River basin from rising.

#### 4.4.2. Estimates of Proportional Contributions of Nitrate Sources in Different Land Uses

As shown in Figure 9b, the contribution ratios of different sources of groundwater nitrate in county, village, and agricultural areas were estimated using a SIAR model. The results showed that the most important sources of groundwater nitrate in different land use types were domestic sewage and manure, and their contribution rates to groundwater nitrate were highest in villages (56%), followed by 44% in agricultural areas and 39% in county areas. This was closely related to the random discharge and stacking of sewage and manure in the village areas. As mentioned earlier, the domestic sewage of most villages drained directly into the Ye River and easily seeped into the groundwater. In addition, the villages in the lower reaches of the Ye River have the habit of raising livestock and poultry, and manure was piled up on the roads of the village. Manure entered the Ye River directly as a result of the scouring effect of the runoff in the flood season and inevitably infiltrated into the groundwater. The contribution rate of soil nitrogen to groundwater nitrate in different land use types (village: 22%; agriculture: 25%; county: 21%) had no obvious difference. The contribution of ammonia fertilizer to the groundwater nitrate in agricultural areas (24%) was higher than that in counties (22%) and village (18%) areas. Similar to ammonia fertilizer, the contribution rate of nitrate fertilizer to groundwater nitrate in the agricultural areas was higher (7%) than that in counties (6%) and villages (3%). It is worth noting that rainfall to groundwater nitrate was highest in county areas, reaching 7%, followed by 3% in agricultural areas and 1% in village areas, which may be due to more serious air pollution that leads to the concentration of nitrate in county areas that is higher than that in agricultural and village areas.

### 4.5. Limitations of This Study and Future Research

In this study, we identified pollution sources and estimated the proportional contributions of nitrate sources to groundwater and river water in the Ye River basin by using a multi-isotope and SIAR model. The results provide a scientific basis for local governments to control nitrate pollution in groundwater and river water. However, there were some uncertainties in the results based on the methods. First of all, we collected the samples of rainfall and sewage, but the end-member values of other sources of nitrate (chemical fertilizer, soil nitrogen, and manure) were based on other relevant studies. This could reduce the accuracy of the source resolution. Therefore, it is vital to determine the end-member values of nitrate pollution sources in future research. Second, we do not distinguish between sources from manure and sewage because their end-member values are duplicated too much. In addition, the δ^15^N-NO_3_^−^ and δ^18^O-NO_3_^−^ values of samples were distributed in the overlapping areas of soil nitrogen, chemical fertilizer, and sewage and manure, which had some influence on the resulting estimation of the proportional contributions of nitrate sources. Thus, in future research, we need to use multiple source tracing methods such as microbial techniques, multiple isotopes (δ^15^N-NO_3_^−^, δ^18^O-NO_3_^−^, δ^37^Cl, δ^11^B, etc.), and multiple statistical methods to accurately identify the sources of nitrate pollution in the water environment.

## 5. Conclusions

Seasonal variations in sources, their transformations, and their proportional contributions to nitrate in the Ye River basin of Hebei Province, China, were elucidated using stable isotopes of NO_3_^−^ (δ^15^N and δ^18^O), water (δ^2^H and δ^18^O), and a Bayesian mixing model (SIAR). Overall, mean NO_3_^−^ concentrations in groundwater (133.5 and 111.7 mg/L in the dry and flood seasons, respectively) were higher than in river water (23.7 and 20.0 mg/L in the dry and flood seasons, respectively). Land use had a significant impact upon the NO_3_^−^ concentration in the Ye River basin, but there was no significant temporal variation in either groundwater or river water. The δD and δ^18^O values of groundwater and river water samples showed that they mainly originated from precipitation in the Ye River basin. The major mode of nitrogen transformation was nitrification for both groundwater and river water. The sources of NO_3_^−^ in groundwater mainly originated from sewage and manure, soil nitrogen, and NH_4_^+^ in fertilizer. However, NO_3_^−^ in river water primarily came from sewage and manure. The results of the SIAR model showed that the primary nitrate sources in groundwater and river water in Ye River basin were sewage and manure. The proportional contributions made by sewage and manure to nitrate in the groundwater and river water were 58% and 48% in the dry season and 49% and 54% in the flood season, respectively. There was little seasonal variation in the contributions of the five nitrate sources in the Ye River basin due to its monsoon climate. In addition, the contribution rate of sewage and manure to groundwater nitrate was the highest in villages (56%), followed by 44% in agricultural areas and 39% in county area. It is worth noting that the contribution of rainfall to groundwater nitrate was higher in county areas (7%) in agricultural areas (3%) and village areas (1%) due to more serious air pollution in the county areas. These results are critical for local governments to develop a priority strategy to control nitrate contamination and achieve water resource sustainability in the Ye River basin.

## Figures and Tables

**Figure 1 ijerph-18-08642-f001:**
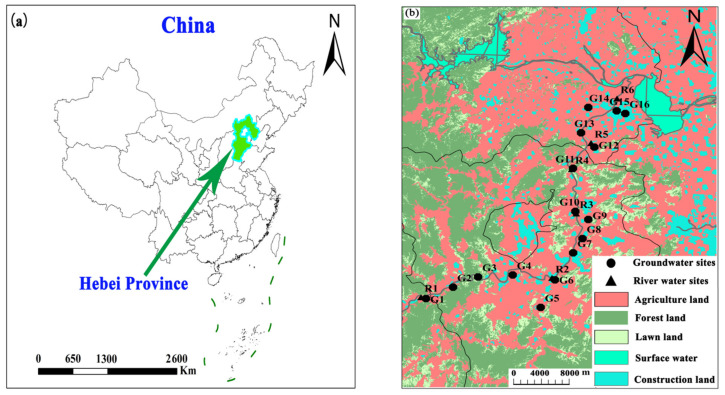
Sketch map of the Ye River basin in Hebei Province, China. (**a**) geographical location; (**b**) sampling sites and land use kinds.

**Figure 2 ijerph-18-08642-f002:**
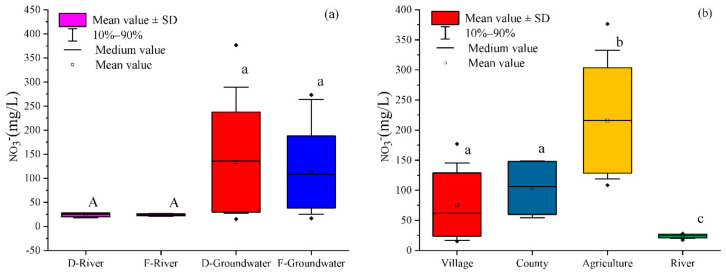
Temporal (**a**) and spatial (**b**) variations in the NO_3_^−^ concentration in groundwater and river water in the Ye River basin. Note: SD = Standard deviation; D = Dry season; F = Flood season; The lines of the SD followed by different lowercase letters for different land use (**b**) designate significant spatial variation at the *p* < 0.05 level by one-way ANOVA.

**Figure 3 ijerph-18-08642-f003:**
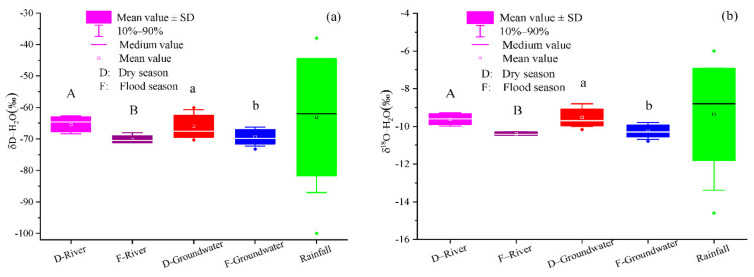
Temporal variation of the δD–H_2_O (**a**) and δ^18^O–H_2_O (**b**) values of groundwater, river water, and rainfall in the Ye River basin. Note: SD = Standard deviation. The lines of the SD followed by different uppercase or lowercase letters for different seasons designate significant temporal variation at the *p* < 0.05 level by one-way ANOVA.

**Figure 4 ijerph-18-08642-f004:**
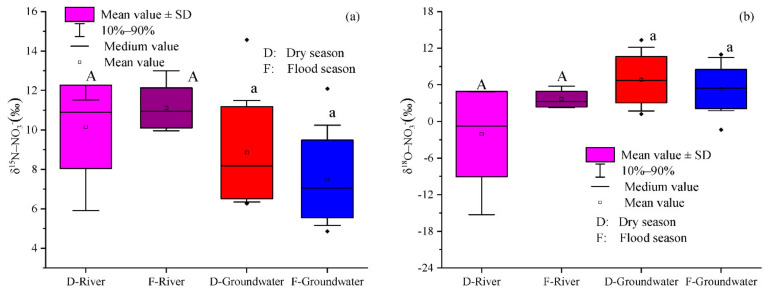
Spatial and temporal variation of the δ^15^N–NO_3_^−^ and δ^18^O–NO_3_^−^ values of groundwater and river water in the Ye River basin. (**a**) Temporal variation of the δ^15^N–NO_3_^−^ values; (**b**) Temporal variation of the δ^18^O–NO_3_^−^ values; (**c**) Spatial variation of the δ^15^N–NO_3_^−^ values; (**d**) Spatial variation of the δ^18^O–NO_3_^−^ values. Note: SD = standard deviation. The lines of the SD followed by different uppercase or lowercase letters for different seasons and land uses designate significant temporal and spatial variation at the *p* < 0.05 level by one-way ANOVA.

**Figure 5 ijerph-18-08642-f005:**
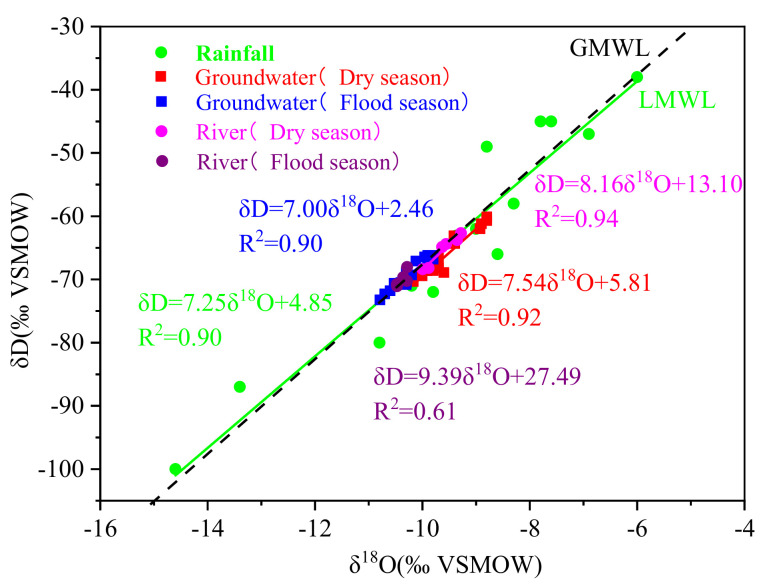
Relationship between δD–H_2_O and δ^18^O–H_2_O in groundwater, river water, and rainwater with respect to global and local meteoric water line (GMWLs and LMWLs) in different seasons in the Ye River basin. Note: Precipitation data were collected from April to October 2018 at IHEG, CAGS.

**Figure 6 ijerph-18-08642-f006:**
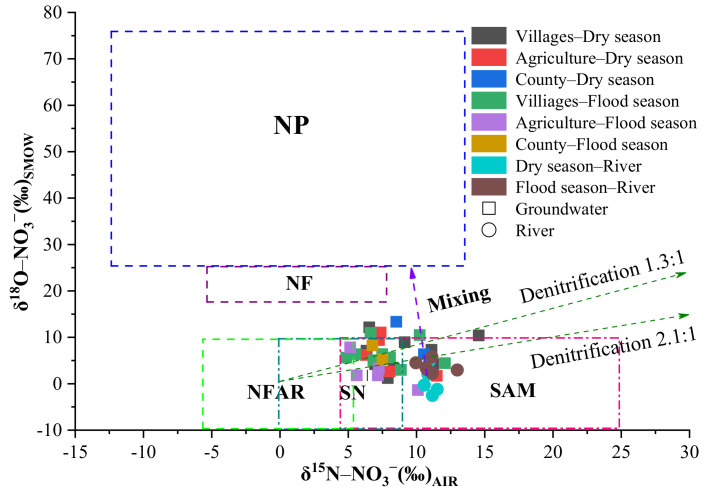
Scatterplot of the δ^15^N–NO_3_^−^ and δ^18^O–NO_3_^−^ values from different seasons and different land uses in groundwater and rivers from the Ye River basin. Note: NP = NO_3_^−^ in precipitation; NF = NO_3_^−^ fertilizer; NFAR = NH_4_^+^ in fertilizer and rain; SN = soil nitrogen; SAM = sewage and manure.

**Figure 7 ijerph-18-08642-f007:**
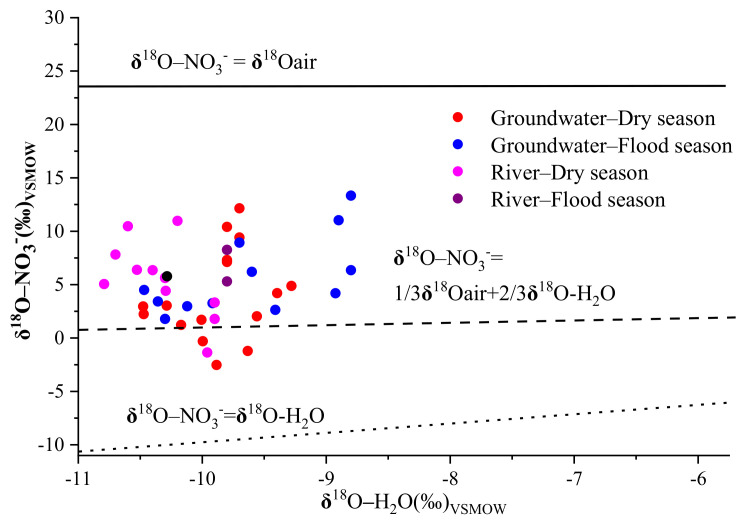
δ^18^O-H_2_O versus δ^18^O-NO_3_^−^ values in different seasons in groundwater and rivers in the Ye River basin. The upper line (blank solid line) indicates that the oxygen atoms in the nitrate produced through nitrification progress are all from dissolved oxygen, the middle line (blank dashed line) indicates that one-third are from dissolved oxygen and two-thirds from surrounding water, and the lower line (blank dotted line) indicates that all oxygen atoms originated from water.

**Figure 8 ijerph-18-08642-f008:**
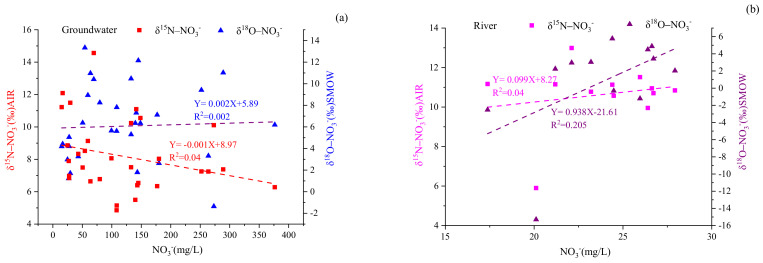
Relationship between NO_3_^−^ and the values of δ^15^N-NO_3_^−^ and δ^18^O-NO_3_^−^ of groundwater (**a**) and river water (**b**) in the Ye River basin.

**Figure 9 ijerph-18-08642-f009:**
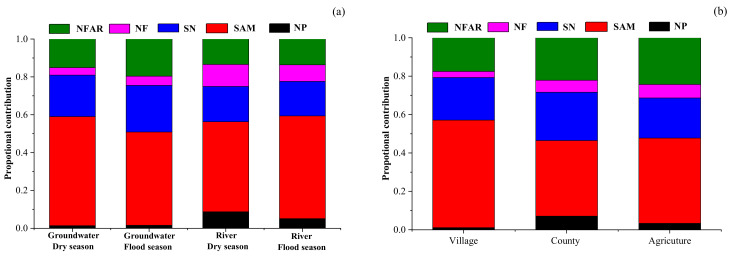
Proportional contribution of five potential NO_3_^−^ sources estimated by a SIAR mixing model in groundwater and river water in the Ye River basin. (**a**) Proportional contributions of NO_3_^−^ sources in different seasons; (**b**) Proportional contributions of NO_3_^−^ sources from different land use types. Note: NFAR = NH_4_^+^ in fertilizer and rain; NF = NO_3_^−^ fertilizer; SN = soil nitrogen; SAM = sewage and manure; NP = NO_3_^−^ in precipitation.

**Table 1 ijerph-18-08642-t001:** Summary of the pH, EC, and DO of groundwater and river water in the different seasons in the Ye River basin.

Parameters	Season	Samples	Range	Mean	SD	Standard
pH	Dry season	Groundwater (n = 16)	6.91–7.72	7.36	0.19	6.5–8.5
		River water (n = 6)	8.05–8.52	8.32	0.18	6.0–9.0
EC (μs/cm)		Groundwater (n = 16)	886–1986	1293	288	—
		River water (n = 6)	1180–1430	1258	96.8	—
DO (mg/L)		Groundwater (n = 16)	2.67–8.79	6.46	1.63	—
		River water (n = 6)	8.13–10.78	9.18	1.16	5.0
pH	Flood season	Groundwater (n = 16)	6.98–8.35	7.44	0.37	6.5–8.5
		River water (n = 6)	8.15–8.37	8.27	0.09	6.0–9.0
EC (μs/cm)		Groundwater (n = 16)	807–2260	1354	406	—
		River water (n = 6)	1003–1113	1058	40.6	—
DO (mg/L)		Groundwater (n = 16)	2.95–9.45	6.86	1.89	—
		River water (n = 6)	7.33–9.78	8.22	0.85	5.0

Note: n = Number of samples; SD = Standard deviation. The standard is the Grade III standard for groundwater and surface water developed by China (GB/T14848-2017 and GB3838-2002, respectively).

## Data Availability

Not applicable.

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
