# Peer review of "Identification of Sources and Transformations of Nitrate in the Intense Human Activity Region of North China Using a Multi-Isotope and Bayesian Model"

_ijerph, 2021, doi:10.3390/ijerph18168642_

Round 1

Reviewer 1 Report

The manuscript by Chaobin Ren et al., is devoted to an important issue -distinguish the characteristics of the spatial and seasonal variation of nitrate in the groundwater and river water in the intense human activity region. Overall, I believe the manuscript is reasonably well written and, in particular, is very detailed in the context of the literature. Basically, I believe the methodology and presentation of the results are sound. However, the figures and tables requires minor revision.

Please check all the figures and tables and be sure that the information’s presented are the same in the titles. Also check the details of every figure (symbols...). I believe that the manuscript must go for another round of revision (figures, tables...).

Please for more details I have mentioned my coments in the document attached. 

Author Response

Thank you for your suggestions. We have revised all the figures and tables according to your suggestions.

Reviewer 2 Report

Dear Authors,

Thank you very much for your hard work and all the time you put into your paper. In this manuscript,  the authors combined the stable isotopes of NO3− (δ15N and δ18O) and water (δ2H and 13 δ18O) with a Bayesian mixing model (SIAR) to identify the sources and transformation of NO3− in 14 groundwater and river in the Ye River basin of north China. According to the results, the authors suggest that the local government should enhance the sewage treatment infrastructure, construct an effective waste storage system to collect manure, and pursue a scientific fertilization strategy (such as soil formula fertilization) to increase the utilization rate of nitrogen fertilizer and prevent the nitrate levels from increasing further. In my opinion, the manuscript is suitable for publication in International Journal of Environmental Research and Public Health, after the authors have addressed the following comments and questions:

Line 140 - 142 : In addition, the bottle is connected with a funnel of 25 cm in diameter, a table tennis ?? is placed inside the funnel to keep the rain from evaporating

Line 142 : “Collect rainwater samples as soon as the rain stops” please rephrase this sentence into the passive form

Line 180 : using ANOVA or MANOVA?

Line 426 – 429 : why don’t this statement is placed in the conclusion part as a suggestion based on the study results

Line 436 – 437 : “please add some explanation about the random discharge and stacking. How can this condition be correlated to the results”

How can distinguish natural soil nitrogen and nitrogen fertilizer in this study

Mention about the limitation and future research is good

Author Response

Line 140 - 142 : In addition, the bottle is connected with a funnel of 25 cm in diameter, a table tennis ?? is placed inside the funnel to keep the rain from evaporating

Response: Yes. We put a table tennis (the diameter of the table tennis is larger than the diameter of the inlet of funnel) above the inlet of funnel which could reduce evaporation of rainwater. When the rain stops, the table tennis can block up the inlet of funnel and reduce the evaporation of rainwater in the sampling bottle. Meanwhile, when it rains, the rainwater can float a table tennis, allowing it to flow into a sampling bottle.

Line 142 : “Collect rainwater samples as soon as the rain stops” please rephrase this sentence into the passive form

Response: We have revised “Collect rainwater samples as soon as the rain stops” as “Rainwater samples were collected as soon as the rain stopped”.

Line 180 : using ANOVA or MANOVA?

Response: In this study, we determined the differences of the NO3 concentration, δ15N-NO3, δ18O-NO3, δD and δ18O-H2O values among different regions and seasons by using one-way analysis of variance (ANOVA).

Line 426 – 429 : why don’t this statement is placed in the conclusion part as a suggestion based on the study results

Response: Thank you for your suggestions. We have put this statement in the conclusion part.

Line 436 – 437 : “please add some explanation about the random discharge and stacking. How can this condition be correlated to the results”

Response: We have added the explanation about the random discharge and stacking.

How can distinguish natural soil nitrogen and nitrogen fertilizer in this study

Response: Nitrogen fertilizer mainly includes nitrate fertilizer and ammonium fertilizer. The δ18O value of nitrate fertilizer was higher than the ammonium fertilizer and soil nitrogen, thus, nitrate fertilizer was easy to distinguish from them. And yet, the δ15N and δ18O values of the ammonium fertilizer (δ15N: from –4‰ to +6‰; δ18O: from –10‰ to +10‰) and soil nitrogen (δ15N: from –3‰ to +9‰; δ18O: from –10‰ to +10‰) partially overlap. However, the mean values of δ15N for soil nitrogen (0.77‰) is more positive than ammonium fertilizer (-2.21‰) based on statistic data from nearly 100 articles. The contribution rate of the soil nitrogen and nitrogen fertilizer to groundwater nitrate pollution can be distinguished by bringing the endmember values of both into the SIAR model, which is a more accurate method at present.

Mention about the limitation and future research is good

Response: Thank you for your recommendation.